



**1** **Feeding the world with soil science: embracing sustainability, complexity**

**2** **and uncertainty**

Pablo Tittonell[1,2,*]
[1]Farming Systems Ecology, Wageningen University, The Netherlands
[2]Natural Resources and Environment, Instituto Nacional de Tecnología
Agropecuaria (INTA), Argentina
*Modesta Victoria 4450, CC 277 (8400) San Carlos de Bariloche, Río Negro,
Argentina

**12** **Abstract**

Feeding a growing and wealthier population while providing other ecosystem
services and meeting social and environmental goals poses serious challenges to
soil scientists of the 21[st] Century. In particular, three dimensions inherent to
agricultural systems shape the current paradigm under which science has to
contribute knowledge and innovations: sustainability, complexity and
uncertainty. The current model of agricultural production, which is also often the
source of inspiration to propose solutions for future challenges, fails at
internalizing these dimensions. It simply does not provide the necessary means
to address sustainability, complexity or uncertainties. Part of the problem is that
these are soft concepts, as opposed to hard goals, and so their definition and
their translation into concrete actions is always subjective. They have to be
sufficiently defined for soil science to embrace them in order to propose viable
solutions to (i) produce food where it is most needed, (ii) decouple agricultural
production from its dependence on non-renewable resources, (iii) recycle and
make efficient use of available resources, (iv) reduce the risks associated with
global change, and (v) restore the capacity of degraded soils to provide
ecosystem services. This paper examines what the concepts of sustainability,
complexity and uncertainty mean and imply for soil science, focusing on the five
priorities enunciated above. It also summarizes and proposes new research
challenges for soil scientists of the 21[st] Century.

**33** **Keywords**: food security; nutrition; global change; soil degradation; agriculture





**1. Introduction**
Feeding a growing and increasingly affluent population while providing other
ecosystem services, and meeting the social and environmental targets of the UN
Sustainable Development Goals (SDG), poses serious challenges to the
management of terrestrial ecosystems used for primary production – and hence
to soil scientists of the 21st Century (Keesstra et al., 2016). Agroecosystems are
complex, dynamic socio-ecological systems in which soils play roles that are
central to their functioning (Walker et al., 2010). They exhibit a number of
properties that can be characterized as structural and dynamic, and are
governed by both biophysical processes and human agency (Conway, 1987). The
design of soil management strategies to meet the SDGs requires knowledge and
innovation, and embracing three dimensions that define the way we regard
agroecosystems nowadays: the notions of sustainability, complexity and
uncertainty. These dimensions describe respectively our aspirations, our
understanding, and the challenges we face towards the future management of
agroecosystems.
The currently hegemonic model of agricultural production issued from the green
revolution, and more recently fuelled by advances in genetic engineering and the
agrochemical industry, appears often as the primary source of a soil scientist's
inspiration to propose solutions for future challenges. This is evident from the
report prepared by the Thematic Group on Sustainable Agriculture and Food
Systems of the Sustainable Development Solutions Network (SDSN, 2013), which
offers new, 'improved' versions of old remedies. In other words, under the optic
of this report, the solution to the problems *caused* by current agriculture
intensification, or to the problems it *fails* to address, are to come from yet further
agricultural intensification. Except that now this is branded as 'sustainable'
intensification. Conspicuous examples of this type discourse are such concepts as
precision agriculture, good agricultural practices, the development of toxin-
producing GM plants supposed to replace pesticides, or more generally the eco-
efficiency and sustainable intensification discourses (cf. Tittonell, 2014a). As
they fail to internalize the mere notions of sustainability, complexity and



uncertainty in agroecosystems these solutions have little chance to help us
achieve the SDG.
Part of the problem is that sustainability, complexity and uncertainty are soft
concepts, as opposed to hard goals. Their definition and their translation into
concrete actions are unavoidably subjective (Meynard et al., 2012). Subjectivity
governs the way in which we perceive and diagnose a problem, as well as the
way in which we come up with solutions. A wrong diagnosis generally leads to
proposing the wrong solution. In many cases the solution to a problem is even
chosen *a priori*, without properly assessing the problem nor the potential impact
of the proposed solution. Blame subjectivity. For example, a well-intentioned soil
scientist working on crop nutrition and soil fertility is often inclined to think that
the solution to food security, or more generally to achieving the SDGs, will likely
come from ample adoption and proper use of fertilisers (e.g. the Fertiliser
declaration of the Abuja Summit, 2006). Yet, after 40 years of such policies in
places like sub-Saharan Africa, and exactly ten years after this new impetus was
installed, we fail to see much progress in terms of agricultural productivity, in
per capita income or food availability in rural areas (UNCTAD, 2013; 2014). Most
conspicuously, and with the exception of very localised examples, there was
weak progress in fertiliser adoption and use in spite of national policies and
large international support to achieve that, while soil fertility continues to
decline.
It is easy to blame policy makers or the social and economic context for lack of
incentives to adoption of technologies, but why not questioning ourselves, soil
scientists, whether the solutions we propose are in themselves real solutions to
actual problems? This challenges us to revisit our research and its contribution
to solving the world's problems from an operational perspective (how do we
make sure that our research is sound and solid?), from an epistemological
perspective (how do we know that we are addressing the right questions,
employing the right methods to approximate the truth?), but also from its
ontological grounds (how do we know what the right questions and relevant
problems really are?). The concepts of sustainability, complexity and uncertainty



in the context of agroecosystems can shed some light on this. They have to be

adequately defined and operationalized for soil science to contribute viable

solutions to five priorities that I deem central to achieving food security:

(i)      produce food where it is most needed;

(ii)     decouple agricultural production from its dependence on non-

renewable resources;

(iii)    recycle and make efficient use of available resources;

(iv)    reduce the risks associated with global change; and

(v)     restore the capacity of degraded soils to support food production and

other ecosystem services.

This paper follows from the introductory paper to this special issue by Keestra et
al. (2016), to examine specifically what the concepts of sustainability, complexity
and uncertainty mean and imply for soil science focusing on the second one of
UN Sustainable Development Goals, Food Security and Nutrition, while
proposing new challenges for soil scientists of the 21st Century.

**17   2. Definitions**

**19   2.1 Sustainability**

Much has been said and written about sustainability (e.g., most recently: James et
al., 2014; Kahle and Gurel-Atay, 2015). Here, I concentrate on the aspects of
sustainability that can help us think through our soils research and its
contribution to food security and nutrition. The most widely quoted definition of
sustainability is that used by the Brundtland Commission of the United Nations
(1987) to define sustainable development as: "[…] *development that meets the*
*needs of the present without compromising the ability of future generations to*
*meet their own needs.*" Yet, in ecology, sustainability has a more strict definition
and means 'the capacity to endure', the ability of ecosystems to remain
productive and diverse indefinitely, bearing a much closer relationship with its
etymology. Sustainability derives from the Latin term *sustinere*, formed by the
particles *sub-* (up from below) and *tenere* (to hold), which resulted in the terms
*sostenir* (old French) and sustain (English). The term sustain appeared in the



western literature since c. 1300 with the following meanings: give support to,
hold up, maintain, endure, bear, undergo, continue, keep up, endure without
failing or yielding, etc. Sustainability is thus the ability to sustain. The adjective
sustainable has been used to mean 'bearable' since 1610s, 'defensible' since
1845, and 'capable of being continued at a certain level' since 1965, when it was
used to describe sustainable growth (cf. Douglas-Harper Etymology Dictionary,

7   2015).

Well-functioning, resilient soils are those able to hold agroecosystems 'up from
below', to sustain their ability to deliver ecosystem services now and in the
future, and to endure the stresses and shocks – anthropogenic or from other
origins – that agroecosystems may be subject to. Policies, management practices
or technological innovations that compromise the ability of soils to sustain
current and future agroecosystems functioning and endurance can be thus
defined as *un*-sustainable. In line with the approach to agroecosystems as socio-
ecological systems, sustainable development *sensu* Brundtland recognizes four
interconnected domains: ecology, economics, politics and culture (Kates, 2011).
Sustainable development implies tradeoffs between objectives derived from
these four domains, and how societies chose between these tradeoffs is central
to sustainability (e.g. Anderson et al., 2015). Let us amend the previous
statement: Policies, management practices or technological innovations that
compromise agroecosystem functioning and endurance, and/or that impact
negatively on the wider societal and environmental objectives, can be regarded
as unsustainable.
There are a few grey zones associated with this definition of sustainability that
deserve further thinking, particularly in the realm of soil science:
1.  *Subjectivity*. Sustainability is a soft concept. It is basically a normative

assertion about choices among societal values. Every social or interest

group will define sustainability and choose its own indicators and

thresholds to make it operational. Is preserving a native forest a

sustainable activity? Is maintaining a living topsoil a sustainable choice?

Is preventing crop residue grazing by livestock in order to restore organic



carbon to soil a sustainable practice? Societies, or power groups within
them, may be ready to trade-off the benefits that can be derived from
these management options in order to satisfy other priorities – to
everyone or just to a few – at any given moment in time.
2. *Boundaries*. Sometimes the causes and consequences of 'sustainable'
decisions are delocalized. This depends on the boundary conditions
chosen to define an agroecosystem. A social group may agree on a certain
management practice or policy that leads to greater sustainability
outcomes within the boundaries of their own ecosystems, but their
choices may result in unsustainable outcomes elsewhere. This not
uncommon in a globalized economy and the examples are plenty. Such
externalities– often overlooked in sustainability assessments – include
aquifer pollution through leaching or the siltation of waterways due to
soil erosion and sediment transport beyond the boundaries of the
agroecosystem.
3. *Thresholds*. As societies trade-off among values to norm on what is
considered sustainable or not, the thresholds necessary to operationalize
these norms are also the result of negotiations and power balances. One
example that relates to soils is the acceptable pollution thresholds defined
for glyphosate concentrations in water. Tolerance levels are set at 700
ppb in the USA, at around 300 ppb in countries like Argentina or Canada
and at 0.01 ppb in Europe. Such broad discrepancies are not the result of
scientists in Europe and the Americas having found different empirical
evidence on Glyphosate toxicity in water. They are just the result of the
capacity for lobbying, negotiation, or weight throwing of different interest
groups in European and American societies.
4. *Objectives*. As the term sustainability is reminiscent of such notions as
maintaining, enduring, etc., it somehow falls short of describing other
important goals of agroecosystem management. Land degradation
assessments based on NDVI trends estimate that about 25% of the
agricultural soils in the word are in a severely degraded state (Vlek et al.,
2008). Much of this is explained by soil degradation (Bai et al., 2008).
Sustainability implies that in many cases it is not enough with





maintaining the *status quo* or 'conserving' soils. Sustainable soil
management may require soil restoration (degraded land) or soil
aggradation (inherently unproductive land) as well – i.e., a sort of
aggradation-conservation agriculture may be needed, cf. Tittonell et al.,

5    2012).

5. *Predictability*. It is hard to foresee what the next generations will need in
terms of resources and ecosystem services. Up until the 19th century salt
was an important asset and even wars were fought about its sources (e.g.
the Wars of the Pacific, 1879-1883). The efficient use and exploitation of
salt would have been a sustainability premise in the past, up until the
moment when other technologies replaced the need for salt. Likewise, it is
hard to foresee the actual consequences of our current soil use and
management for the future generations. How do we know if a current
disturbance will lead to a future problem or a future advantage? Think of
the high cultural value assigned today to formerly degraded landscapes,
such as the dune landscape in north-western Europe that originated after
massive peat extraction from the soil surface, or the high economic value
of okoumé (*Aucoumea klaineana*) trees in Equatorial west Africa, that
grow in places that were cleared through slash and burn agriculture in
the past.
One may well imagine that healthy soils and ecosystems will be as necessary for
food production in the future as they are today – as they have been for over 7
millennia – but it might well be that the prime functions and services they are
expected to provide will be other than those of today. These grey zones pose
important ontological questions to soil scientists of the 21ts century. Yet they
should not be used as arguments to disturb ecosystems beyond acceptable
thresholds, which must be defined using our best available scientific knowledge.
It makes sense to be conservative when it comes to choosing thresholds to assess
sustainability, particularly in the case of practices or technologies that may have
long term impacts on society and the environment that are yet unknown or
poorly understood. The goal of sustainable development is to propend towards a
form of intergenerational equity, maintaining the ability of global ecosystems to
sustain future generations.



**2.2 Complexity**
We take for granted that agroecosystems are complex yet they are often a
simplified version of a natural ecosystem. Through agriculture, we have
simplified natural ecosystems so that the flows of solar energy, water and
nutrients can be steered in the narrow direction of producing selected forms of
plant and animal biomass.  Complexity was inevitably lost in the process.  When
assessed through indicators such as heterogeneity, diversity, dynamics and
feedbacks, we can attest dramatic losses in structural, organisational and
functional complexity of ecosystems through agriculture. A conspicuous example
would be the loss of structural and functional complexity of soil biodiversity
when forest soils are put under grazing or cultivation (e.g. Lupatini et al., 2012).
The human dimension of agroecosystems, however, introduces new forms of
complexity that are hard to unravel using ecological knowledge and principles.
The actual complexity inherent to agroecosystems can only be understood when
they are perceived as social-ecological systems.
Complexity can be seen as the mere number of components and interrelations in
a system, or defined by the nature of such relationships, or by the emerging
properties associated with them. Complexity can be associated too with the level
of knowledge we have about the functioning of a system: less understood
systems tend to be considered more complex.  In soil science, complexity has
been most often studied in the various fields of soil biology, notably through the
study of soil trophic networks, but also in the study of the physicochemical
properties, functioning and spatial arrangement of the minerals and particles
that define soil composition and structure. More recently, the notion of
complexity and ways of assessing it have been applied in the study of the close
links between soil management, spatial patterns and collective human decisions
or social networks (Rufino et al., 2011; Isaac, 2012; Tittonell, 2014b; Andrieu et
al., 2014; Baudron et al., 2014; Speelman et a., 2014). The next examples will
focus on these aspects of social-ecological systems, as they bear a close



relationship with the challenge of achieving the UN Sustainable Development
Goals through soil science.
*Collective action and decision-making*
Introducing a technology or a set of soil management principles in a rural
community is often a challenge. Lack of adoption is generally ascribed to the
communities' poor understanding of the benefits of the technology, to their lack
on interest in or need for it, or to the absence of economic, social or policy
incentives to implement it. Often the problem is much more complex than that. In
Malawi, soil and water conservation measures have been profusely promoted to
halt soil degradation and increase agricultural productivity in the recent past.
Some communities were more prone than others to adopt them. In-depth social
studies in some of these communities of adopters and non-adopters revealed
that the topology of the local social networks that characterised each community
was closely associated with the level of adoption of soil and water conservation
measures proposed to them (Khonje, 2012; Figure 1).



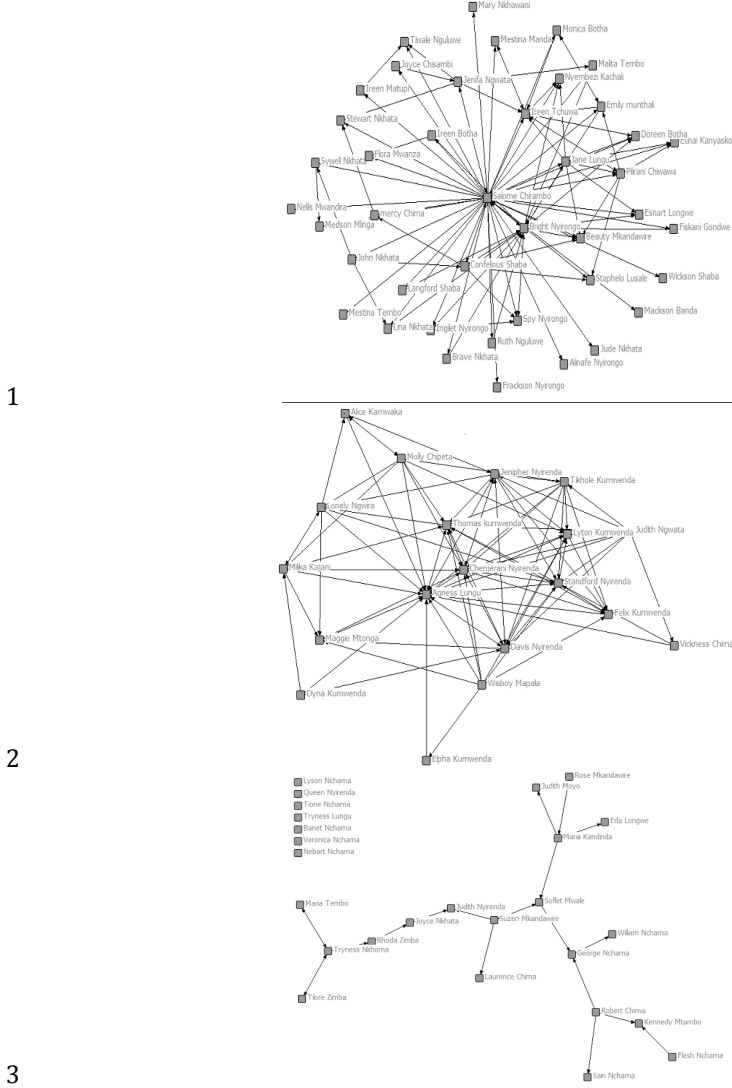

Figure 1: Social network topologies of three villages in Malawi: (A) Ziwambera; (B) Kanyinbwa;
(C) Kamkondo (Source: K. Khonje 2012, Wageningen University).
It must be recalled that several soil and water conservation measures, such as
building terraces, planting tree lines or digging ditches, require collective effort
by members of the community, especially when the landscape is fragmented into
very small properties (of often less than half a hectare). The community of
Ziwambera village, where the adoption of soil and water conservation measures





was successful, exhibited a traditional type of social organisation, in which it is
possible to evidence the strong leadership of a local chief, who relates to the rest
of the households often through family ties (Fig. 1A). Kanyimbwa village social
network reveals a situation in which more than one strong node of relations
exists, resulting in a more complex and diverse communal decision-making
model (Fig. 1B). Kamkondo village social network points to a loosely connected
society (Fig. 1C), composed largely by migrating families with rather weak ties
amongst them, with unconnected households, with ties to nearby townships as
well, and where community action is hard to coordinate. Non-surprisingly, soil
and water conservation measures that required collective effort were least
adopted in the latter community. Social network structures can also enhance or
hinder the flow of information amongst members of the community and very
strongly influence adoption (Khonje, 2012).
*Resource flow networks*
At farm level, complexity is evident in the way diversity is organised.
Diversification of farm activities *per se* brings about greater resilience and
adaptability in the face of external shocks, such as low or volatile prices, a
drought or a pest outbreak. This is due risk spreading, or the classical egg-in-
different-baskets effect. Yet diversity *per se* is not enough to grant efficiency in
resource use, as observed for example amongst family dairy farmers in Mexico
by Cortez-Arriola et al. (2014) or in rice-based systems of Bangladesh by
Arvidashkan et al. (2015). To be functional to a given objective, diversity must be
organised in a certain way. In ecology as well as in agriculture, the relationship
between biodiversity and efficiency, for example nutrient use efficiency, is
generally elusive, and it may become evident only when moving across scales.
Processes deemed inefficient at one scale may contribute to greater system
efficiency at higher scales (van Noordwijk and Brussaard, 2014). The analysis of
individual farms as ecological networks in crop-livestock agroecosystems of East
and southern Africa revealed however that resource (nitrogen in this case) use
efficiency was closely associated with the complexity and organisation of the
internal network of resource flows (Rufino et al., 2009; Alvarez et al., 2012).
Across household types and climatic regions, N use efficiency and food self-



sufficiency at farm level were better explained by the nature of the resource flow
network complexity, in the form of path organisation and internal recycling, than
by the total amount of externally-sourced N entering and flowing through the
farm system annually (Table 1).

Table 1: Indicators of resource endowment, and of the size and organisation of the network of nitrogen flows within eight case study smallholder farms (from: Rufino et al., 2009; Alvarez et al., 2012)

| Location/ Farm type | Cropped land (ha) | Livestock owned (TLU) | Farm N network size | | | Farm N network organisation | | Farm N use efficiency (kg kg N⁻¹) | Food self sufficiency ratio |
|---|---|---|---|---|---|---|---|---|---|
| | | | Total system throughput | Dependency on imports (%) | Finn's cycling index (%) | Average mutual information | Diversity of flows | | |
| Ethiopia | | | | | | | | | |
| Poorer | 0.3 | 1.2 | 230 | 72 | 2.9 | 1.1 | 2.2 | 23 | 0.4 |
| Wealthier | 2.4 | 10.0 | 1340 | 66 | 2.6 | 1.3 | 2.4 | 18 | 1.7 |
| Kenya | | | | | | | | | |
| Poorer | 1.0 | 0 | 45 | 45 | 2.2 | 1.1 | 2.5 | 74 | 0.3 |
| Wealthier | 2.9 | 3.5 | 190 | 34 | 11.0 | 1.7 | 3.3 | 216 | 1.2 |
| Zimbabwe | | | | | | | | | |
| Poorer | 0.9 | 0.3 | 40 | 65 | 0.9 | 1.0 | 2.2 | 44 | 0.5 |
| Wealthier | 2.5 | 5.4 | 480 | 45 | 5.5 | 1.5 | 2.9 | 86 | 3.4 |
| Madagascar | | | | | | | | | |
| Poorer | 2.7 | 3 | 110 | 33 | 3.5 | 1.2 | 2.6 | 122 | 1.9 |
| Wealthier | 6.9 | 12 | 400 | 31 | 2.5 | 1.4 | 3.4 | 198 | 4.7 |

Total system throughput is the sum of all N flows between all components (activities) of the farming system, expressed here in kg N per family member to allow for comparisons across farms of different size; Dependency on imports is the ratio between N flows into the farm system and total system throughput; Finn's cycling index is calculated as the ratio of the sum of all internal flows to total system throughput; Average mutual information (AMI) is the average number of connexions of each system component and the diversity of flows (HR) or statistical uncertainty is the maximum number of possible connexions between components, or the upper limit to AMI; both AMI and HR are measured in bits (binary decisions); if all the components of a system are connected and the total flow is equally distributed among all components, AMI will approach zero; typical values of AMI in natural ecosystems range between 0 and 6; Farm N use efficiency is the ratio of total biomass productivity to total N flowing into the system; Food self-sufficiency ratio is the ratio of edible calories produced on farm to caloric household needs.

*Spatial patterns*
At field scale, soil complexity is often reflected in its spatial heterogeneity, which
is both inherent to soils and also the result – intentional or not – of human
agency.  I have worked extensively on the study of spatial soil heterogeneity, its
causes and consequences, in smallholder African agriculture (e.g. Tittonell et al.,
2005a,b; 2008; 2010; 2013; 2015; etc.). Figure 2 summarises schematically the
three most common patterns of soil heterogeneity that can be found across sub-
Saharan Africa. Fig. 2A represents the somewhat classical ring effect documented
by Prudencio (1993) in Sudano-sahelian zones of West Africa, which describes a
gradual decline in soil fertility at increasing distances from the homesteads,
organised in villages or *hameaux*. Fig. 2B illustrates discrete patterns of spatial
heterogeneity in highly populated and fragmented landscapes, in which
homesteads are not centralised in a village but located within individually owned
farms, and where soil fertility declines at increasing distances to them (e.g.
Tittonell, 2003). One of the most common patterns in less populated regions of
East and southern Africa is the discontinuous gradient, which strictly speaking is
not really a gradient (Fig. 2 C). In these patterns, described in Zimbabwe by e.g.
Carter and Murwira (1995), village fields and outfields are not necessarily
contiguous as in the West African case. Due to the rapid increase in population




densities that can be seen in rural Africa the first and the last pattern are
gradually disappearing, tending towards more fragmented, spatially
heterogeneous situations as that illustrated in Fig. 2 B.

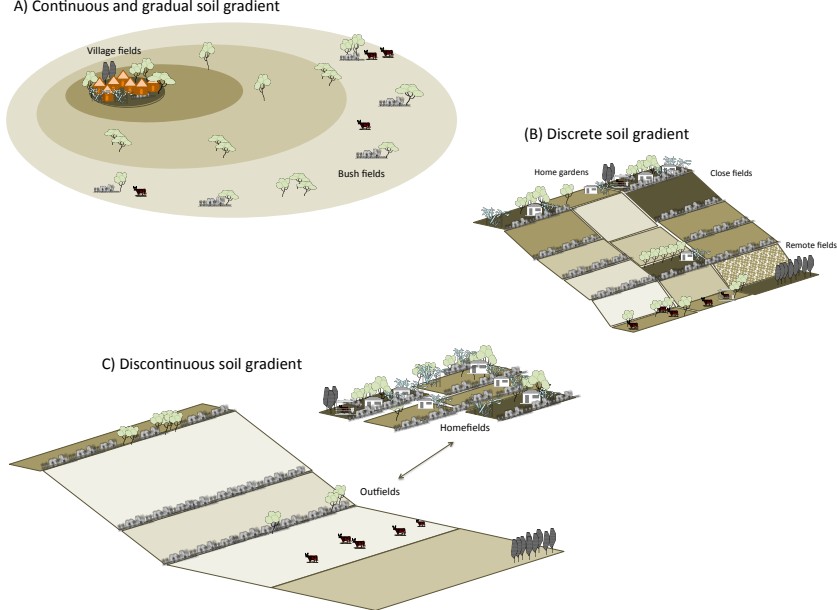

Figure 2: Schematic representation of common spatial patterns of soil heterogeneity in
smallholders farming systems of sub-Saharan Africa. Modified from Tittonell et al., 2015.
Spatial soil heterogeneity is important because it is associated to a variable
extent with heterogeneity in crop productivity, in their response to nutrient
additions, or in the performance of soil improving technologies (Vanlauwe et al,
2006; Wopereis et al., 2006; Zingore et al., 2007; Fermont et al., 2009; Tittonell
and Giller, 2013; Bruelle et al., 2015; Diarisso et al., 2015). Fertiliser
recommendations tend to fail in sub-Saharan Africa due to this mere reason. The
agronomic and economic efficiencies of fertiliser use can vary from benefit to
failure within a single, spatially heterogeneous farm.  N-fixing legumes may not
be able to grow – let alone to fix N – in the poorest or degraded fields of small
heterogeneous farms (Giller et al., 2011).  But spatial heterogeneity is also
important because in some cases it is the result of farmers' intention. Farmers
may create 'islands of soil fertility' by concentrating resources in small portions



of land, where yields are secured through more intense crop husbandry, early
planting and nutrient additions from diverse sources (Henry et al., 2009;
Castellanos-Navarrete et al., 2014; Diarisso et al., 2015). Such fields tend to
exhibit greater soil organic matter, nutrient availability and water holding
capacity, and a great diversity of plant species grown in association, which
contribute to household nutrition through diverse diets (Figueroa-Gomez et al.,
2008). These fields can be also seen as a bank or an insurance asset to secure
food production in years of scarcity or erratic rainfall, thereby contributing to
whole-system resilience (cf. Tittonell, 2014b).
The above examples indicate that agroecosystem complexity is not scale-
agnostic. Understanding complex social relations at village or landscape level,
complex networks of resource flows at farm scale, or complex patterns of natural
or human-induced soil spatial heterogeneity at field scale are central to the
design of soil management strategies towards the achievement of the SDG. The
approaches to be used in soil science must therefore embrace scale-dependency.
A classical way of dealing with scales and integration levels in soil science, and
agricultural science in general has been through the model of hierarchical
confinement (e.g. Fresco and Westphal, 1988). Although this model became
obsolete after the notion of *panarchy* emerged in ecology (Gunderson and
Holling, 2001), it is still widely applied in soil science and must be revised in the
light of new insights from complex systems theory (Vandermeer and Perfecto,
2013) – another white glove lying on the ground of soil scientist of the 21[st]
century.
**2.3 Uncertainty**
Uncertainty in agroecosystems is associated with the probability of occurrence
of a certain event and with the sensitivity to and type of response to this event
that may be expected. Events are normally associated with the behaviour of
external driving variables, such as market or climatic ones, but also with a
systems' own structure and function. Uncertainty may arise from stochastic
processes and/or unforeseeable futures, but also from our inability to yet



understand how things work in nature. There are thus cases in which the type of
response to be expected is not known *a priori*, and there are others in which
possible responses are known but their patterns, in terms of extent or intensity
are not known. These two types of cases define two different domains of
research questions relevant to soil science and agriculture. The first one, the
uncertainty about the unknown, relates to science's continuous quest to unravel
reality and how it works. The second one refers to the probability of occurrence
of a known response, which in agriculture often boils down to risk assessments.
Climatic risks and price risks are inherent to agriculture, and proper soil
management may help curtail them. But the risk of not obtaining the expected
response to a given investment is a serious matter for resource-constrained
farmers, for whom the consequences of a years' failure may be felt over more
than one season. I will explore through a few examples the implications for soil
science of uncertainties about the unknown and uncertainties about responses,
recognising that there is still a third domain of research questions not addressed
here that is concerned with how to reduce uncertainty in agroecosystems.
*Uncertainty about the unknown*
In the 1940s a group of competent toxicologists led by William B. Deichmann
conducted a number of thorough studies using state-of-the-art methods to
conclude that the active ingredient dichloro-diphenyl-trichloroethane, or DDT,
could be safely released to the environment for its use as insecticide. DDT was
one of the first wide spread synthetic pesticides, and its widespread use led to
resistance in many insect species. In the early 1970s, a paper authored by
Deichmann (1972) himself and other studies provided enough evidence for the
US Environmental Protection Agency to finally forbid the use of DDT as it became
known to be toxic to humans, persistent in the environment, travel long
distances in the upper atmosphere, and accumulate in fatty tissues of living
organisms. What did actually happen between the 1940s and the 1970s? Why
was DDT first considered innocuous or degradable and 30 years later banned
and labelled as poisonous for humans, wildlife and the environment?
There are several possible answers to these questions. In the fist place, the
ecotoxicity of certain chemicals when applied in small doses may only appear




through cumulative effects (cf. http://www.efsa.europa.eu/fr/node/872721).
Time is needed for problems to arise, or to become evident. Second, and most
importantly, the capacity of science to detect the adverse effects of a certain
molecule released to the environment can progress substantially in 30 years.
Problems that were overlooked or remained undetected in the past could be
later on well understood and documented. (And the amount of scientific
evidence that needs to be accumulated to be able to bend the arm of the chemical
industry in court cases is not a minor detail). Examples such as this one should
teach us about the long-term risk (uncertainty) associated with the widespread
release of synthetic molecules, either as chemical formulations or produced by
genetically modified organisms (cf. Cheeke et al., 2012), into the environment.
Alarming ideas such as the commercial release of genetically engineered
microorganisms for soil amendment have been underway for a while (e.g.
Viebahn et al., 2009), with unknown consequences for soils and the environment.
The adverse effects of certain soil management practices will only arise once a
certain sustainability threshold is crossed. This is the case, for example, of
turbidity regimes in shallow lakes as affected by the continuous charge of
incoming nutrients in regions of intensive crop and animal agriculture (e.g.,
Scheffer et al., 2001). Such thresholds are often termed 'tipping-points'
associated with non-linear, irreversible or hardly reversible dynamics that
describe multiple possible equilibriums (e.g., Walker et al., 2010). Evidence of
such complex dynamics in agroecosystems is analysed in Tittonell (2014b),
where household dynamics and their livelihood strategies are portrayed as
complex systems that follow non-linear trajectories, with impacts on soil
productivity and spatial heterogeneity. Lower equilibrium situations, in which
farms are caught within vicious cycles of soil fertility depletion, low productivity,
meagre incomes and poor investment capacity to restore soils are often termed
poverty traps (Tittonell and Giller, 2013). Further evidence on the dynamics of
smallholder African households and their impact on soils was presented by
Valbuena et al. (2015), who suggested that they often behave as 'moving targets'
for soil research for development.



*Uncertainty about response patterns*
Why would farmers invest in actions to restore degraded soils if they do not
know the possible outcomes of such actions? The dynamics of soil
responsiveness to restorative measures can be characterised by two attributes:
hysteresis and time lags (Tittonell et al., 2012). Hysteretic responses are
desirable, so that the trajectory of soil recovery diverges as much as possible
from the one of degradation, and soils may in time recover properties that are as
close as possible to those exhibited by non-degraded soils (Figure 3). In a long-
term study in Benin diverse soil management practices, from a control without
soil amendments and crop residue removal, to a full treatment with application
of mineral and organic fertilisers and crop residue incorporation to soil, were
compared for 20 consecutive years. After 10 years, however, soil in the control
treatment degraded to such an extent that maize yields became virtually nil (red
markers in Fig. 3B). During the next 10 years, in which these soils received
amendments of organic and mineral fertilisers and of crop residues at the same
level as the full treatment (blue markers in Fig. 3B) their productivity recovered,
but never reached the level recorded in soils that received such inputs
throughout the 20 years of the experiment (green markets in Fig. 3B); the
relative yield was about 60%. In some cases, although hysteretic responses are
achievable and soils can potentially be restored, the fist signs of response to
restorative measures only show up after a number of years of farmers'
investment (e.g. Corral et al., 2015). Such time lags often deter farmers'
investments in soil restoration measures.  Note now in Fig. 3B that when the
degraded soil started receiving organic matter and nutrient inputs it
immediately produced a yield that was comparable with the initial yield without
inputs, and it took four years before a substantial response was obtained.



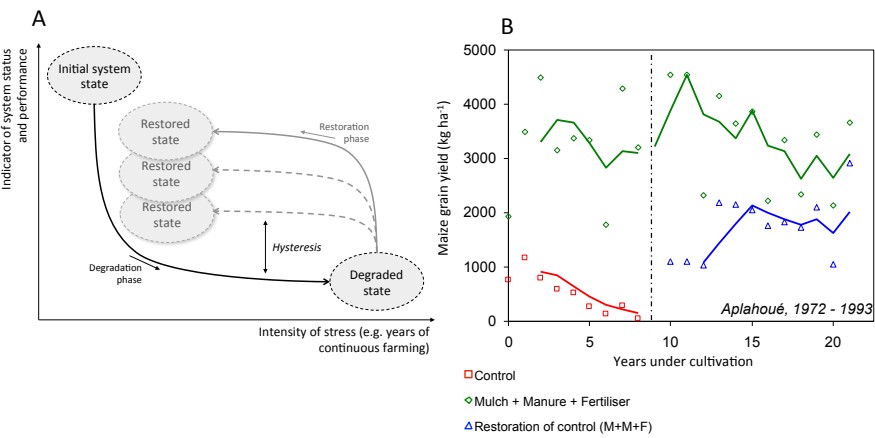

Figure 3: (A) Schematic representation of hysteretic responses to sol rehabilitation in which
three alternative 'restored states' are represented; (B) Maize yields without nutrient inputs and
crop residue removal (control) and with additions of manure, fertiliser and crop residues from
the beginning or after year 10 (restoration of control) in a long term experiment in Aplahoué,
Benin. Source: *Institut National de Recherches Agronomiques du Benin* (INRAB).
Farmers perceive uncertainty in responses mostly in the short term. Farmers
engaged in a 4-year development project supporting sustainable agricultural
intensification practices in 3600 upland rice fields in Madagascar were trained
and encouraged to adapt conservation agriculture and nutrient management
practices as they saw fit (Bruelle et al., 2015). As initial responses to synthetic
fertiliser applications were highly variable, with yields ranging from 0.4 to 4.2 t
ha$^{-1}$ on the most risky hillside fields, farmers adopted conservative fertiliser
rates, intuitively targeting the highest efficiency zone of a fertiliser response
curve. Farmers' fertiliser N rates on hillside fields (n = 857) were in the order of
20 to 35 kg ha$^{-1}$ on average. The variability in crop responses observed was large
and multivariate, which is a common phenomenon observed on smallholder
farms (e.g. Vanlauwe et al, 2006; Wopereis et al., 2006; Zingore et al., 2007;
Tittonell et al., 2008; Fermont et al., 2009; Diarisso et al., 2015). The analysis of
yield variability using classification and regression trees revealed that significant
yield responses to N fertilisers (>15%) were only observed in in 55% of the
fields and only in two out of four years, with a cut-off threshold of 28 kg ha$^{-1}$ N
applied (cf. Bruelle et al., 2015). Amongst the fields that received less than 28 kg





ha$^{-1}$, 13% of them exhibited higher yields than the average of fields receiving
greater fertiliser rates. It is not surprising that such uncertainty in crop
responses to synthetic fertilisers deters resource-constrained farmers from
investments in supposedly 'quick fixes' for soil fertility management.
**3. Soil science priorities for food security and nutrition (SDG 2)**
In the introduction to this paper I proposed five priority areas that soil science
needs to address in order to contribute to food security and nutrition (SDG 2):
(i)      Produce food where food is most needed;
(ii)     Decouple agricultural production from its dependence on non-

13           renewable resources;

(iii)    Recycle and make efficient use of available resources;
(iv)    Reduce the risks associated with global change; and
(v)     Restore the capacity of degraded soils to provide food and ecosystem

17           services.

It takes much more than just good soil management to achieve food and
nutritional security and addressing these priorities will often exceed the realm of
soil science and call for trans-disciplinary research. Non-exhaustively, Table 2
provides some examples of pathways that could be followed and possible
research approaches to address these priorities from the perspectives of
sustainability, complexity and uncertainty.

Table 2: Soil research priorities to achieve UN Sustainable Development Goal 2 Food Security and Nutrition, and examples of approaches from the perspectives of sustainability (what needs to happen), complexity (what do we know/ should consider) and uncertainty (foresee and minimise what could go wrong)

| Priority | Sustainability | Complexity | Uncertainty |
|---|---|---|---|
| Produce food where food is most needed | Design food production systems and soil management technologies adapted to the context of resource-poor, less favourable, or less developed regions;<br><br>Think beyond major staple foods to design | Transdisciplinary understanding of institutional, societal (cultural) and ecological interactions in resource-poor, less favourable or less developed regions;<br><br>Understanding and exploiting the role of | Make food production systems more resilient and adaptable to shocks and stresses through diversification, and credit and insurance policies to assist in case of natural and human-made disasters; |





| | diverse, nutrition-sensitive landscapes;<br><br>Promote local autonomy and belligerence in food systems governance; | man-made 'fertility islands' in agroecosystems and other forms of indigenous precision agriculture; | |
|---|---|---|---|
| Decouple agriculture from non-renewable resources | Intensification of food production systems through diversification and ecological replacement;<br><br>Foster the provision of soil-mediated ecosystem services (e.g. nutrient regulation, biological N fixation) | Understanding of soil trophic networks and biodiversity, and their contribution to nutrient capture and cycling;<br><br>Understanding to what extent N fixation by organisms other than legume plants contribute to ecosystem's N budget (e.g. in grasslands); | Foster the provision of ecosystem services to reduce pest impacts (e.g., pest regulation, soil suppressiveness);<br><br>Minimize agriculture's dependence on fossil fuels and volatile oil markets; |
| Recycle and make efficient use of available resources | Look beyond partial efficiencies at plot/crop scale to embrace whole-system efficiency at farm and landscape levels;<br><br>Improve the recycling of energy and matter within and between agroecosystems, and between rural and urban areas; | Efficiency is not scale-agnostic: new theory is needed, perhaps ascendency (cf. Ulanowicz, 2001), to measure systems' efficiencies at farm and landscape level;<br><br>Understanding the links between biodiversity and efficiency at different scales; | Introduce the notion of risks in the calculation of efficiencies;<br><br>Can risks of leakages be minimized through recycling?<br><br>What are the tradeoffs associated with pursuing efficiency at one single scale/component? |
| Reduce risks from global change | Foster the provision of soil-mediated ecosystem services (e.g. water regulation);<br><br>Ecological intensification of food production systems through diversification and exploitation of niche-complementarities (e.g. crop-livestock-tree systems); | Understanding the relationship between functional soil biodiversity and its adaptive capacity in the face of climate change;<br><br>Analysis of nature's structure-function relations and adaptability mechanisms to inform agroecosystems design (e.g. ligneous components in grasslands); | How can soil act as a reactor to detoxify the environment?<br><br>What are meaningful indicators and tipping points to monitor soil functioning and service provision?<br><br>How can soil information best contribute to insurance policies and mechanisms? |
| Restore degraded soils | Reduce time lags and initial investments;<br><br>Whenever possible, restore soils while producing;<br><br>Make use of locally available resources and knowledge for soil restoration;<br><br>Promote local autonomy and ownership in soil restoration initiatives; | Understanding of mechanisms behind hysteretic and non-hysteretic responses to soil restoration;<br><br>Understanding of sequential processes in soil and associated functions during a restoration phase. | What are the costs and benefits associated with restoring soils?<br><br>How can smallholder farmers overcome time lags?<br><br>What are the institutional and governance mechanisms necessary to ensure long-term engagement in soil restoration? |





In the light of sustainability, complexity and uncertainties, let us think critically
about the model of agricultural intensification that several international
organisations – with good intentions – are pushing in developing countries: what
is the reasoning behind the idea that a mono-culture of a uniform maize hybrid
receiving large inputs of synthetic fertilisers and other agrochemicals is the most
appropriate cropping system for a subsistence, sometimes illiterate household in
a remote, food- and land-insecure, and inaccessible rural village exposed to
climatic risks and demographic pressure, and located in a fragile ecosystem in
the tropics, where input prices are ten times higher than those paid by farmers in
e.g. Europe? (On input prices cf. Tittonell et al., 2007). Can such simplified and
risk-prone cropping system really contribute to food security, namely
availability, access, stability and utilization of food, and nutritional diversity in
smallholder contexts?  Beyond any ecological consideration about the
consequences of simplifying tropical ecosystems in this way, and just from a
mere economic perspective, what would happen with rural food security and its
sustainability under the uniform hybrid scenario if for example oil prices (and
fertilisers) sky rocket, or if multinational seed companies manage to enforce
payments for intellectual property upon farmers as is happening in other parts of
the world?
Sustainability, complexity and uncertainty represent respectively our
aspirations, our understanding, and the challenges we face towards the future
management of agroecosystems to achieve the UN Sustainable Development
Goals. Although several epistemological questions remain unanswered around
these notions, particularly around sustainability as a guiding concept, they
represent three complementary perspectives from where we can analyse the
potential of our contributions from soil science. On the basis of this, here are
three lines of action that may help us soil scientists of the 21st Century make our
research more impact-oriented towards the achievement of food security and
nutrition:
1. Let us invest effort and creativity in 'translating' the new insights and

33       understandings coming from soil science into knowledge-intensive



innovations that can contribute to food security. Exciting research results
on formerly unknown soil interactions in the root zone are a sure ticket to
a high impact publication. Yet the path from understanding processes in
soil to implementing this knowledge in the design of sustainable
agroecosystems is a long and tortuous one. We can contribute to make it
shorter. But doing research on applied science, moving from questions
such as '*how does this work*' (analysis) towards '*how to make it work*'
(design) penalises soil scientists, as design-oriented research is hard to
publish in high impact journals. This requires also to rethink the scientific
reward systems in research and the academia;
2. Let us dare to challenge the establishment. Many of us in soil science are
under the influence of those who were our mentors, who provided
guidance and/or trained us as researchers. They are the ones who
nowadays act as consultants for international organisations, who direct
research institutes, participate in boards, act as journal editors, evaluate
projects, etc. Most soil scientists of that generation were educated and
developed their careers within the paradigm of the green revolution:
improved germplasm, fertilisers and agrochemicals can fix anything.
Some scientist of that generation knew how to adapt and change their
views and discourses as new evidence and methods came along. For
others, a change of view at this point is felt in a way as being equivalent to
accepting that what they did in the past was wrong. Let us engage in
persuasive actions to break such inertia, and let us not be shy when it
comes to defending our new insights – provided that they are supported
by sound scientific evidence;
3. Let us contribute to creating credible narratives that bring important
messages to society and/or install useful ideas among agroecosystem
managers, policy and decision-makers. Narratives are well-packed pieces
of information that convey simple, common sense messages. A popular
one is this: "*it will be impossible to feed the world without GM crops*".
Although this assertion does not resist any serious scientific scrutiny, the
industry managed to install this message successfully amongst some
policy makers, the public opinion in general and, most disquietingly,



among certain members of the scientific community. I propose to liaise
with communication experts to develop simple narratives to convey
meaningful messages from soil science to society, with the ultimate
intention of influencing policy and research agendas;
The complexity of the social-ecological systems where food is produced and
consumed requires trans-disciplinary research, knowledge brokerage, and a
permanent dialogue with policy makers. The relationship between scientific
evidence and policy development and implementation is not rectilinear, but a
rocky path (Caceres et al., 2016; Keesstra et al., 2016). It would be naïve to think
that we can feed the world just with soil science, as the title of this paper
ironically claims. Hunger is not the result of insufficient agricultural production.
Hunger is the result of poverty and inequality. Feeding the world, or achieving
the second of the UN Sustainable Development Goals, requires much more than
soil science and it certainly exceeds agricultural research. Yet it helps to be
aware of the role we can play in this puzzle, a role that is not minor.

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
