# Peer review of "Published: 9 March 2016"

_SOIL, 2016_

## Referee Comment (RC1) · Anonymous Referee #1 · 13 Apr 2016

General comments: The manuscript provides a timely and thorough reflection of soil science limitations to address important challenges in the search for sustainable development trajectories, that are holistic enough to capture the complexities associated with context diversity, and that recognize and effectively adapt to an increasingly uncertain future. It is argued that an important limitation to embracing the sustainability, complexity and uncertainty dimensions in current agricultural scientific enquiry is largely driven by the failure of defining them in a way that facilitates linking knowledge to action. The author rightly proposes a pragmatic approach that uses five tangible goals to guide the internalization process of these three dimensions into the design of soil management strategies to meet the SDGs, namely, (i) producing food where it is most needed, (ii) decoupling agricultural production from its dependence on non-renewable resources, (iii) recycling and making efficient use of available resources, (iv)

[Figure]

reducing the risks associated with global change, and (v) restoring the capacity of degraded soils to provide ecosystem services. Soil scientists are further challenged to be more critical and encouraged to examine the relevance, credibility and legitimacy of their research contributions to solving the world's problems. In order to contribute to fill gaps identified the author thoroughly reviews the definitions of sustainability, complexity and uncertainty. This is followed by a systematic examination of soil research priorities to achieve the SDG Goal #2 Food Security and Nutrition that clearly links each of the five tangible goals described above with approaches to embed the perspectives of sustainability (what needs to happen), complexity (what we know/should know) and uncertainty (foresee and minimize what could go wrong) into future soils research efforts. This is an excellent thought-provoking manuscript of broad international interest that elegantly articulates challenges and opportunities for soil scientist in the 21st century. It makes a needed call for reflection and critical thinking in future research to ensure that soils research is intimately linked to relevant development problems thus effectively contributing to achieve the SDGs in general and the SDG Goal #2 Food Security and Nutrition in particular. I fully recommend the manuscript for publication in the SOIL journal and have included a few suggestions for consideration.

Scientific questions/issues: P3 L8-11 Suggest including the following reference to support the statement in this sentence:

Prell C., Hubacek K., Reed M., Quinn C., Jin N., Holden J., Burt T., Kirby M., Sendzimir J. 2007. If you have a hammer everything looks like a nail: traditional versus participatory model building. Interdisciplinary Science Reviews 32(3): 263-282.

P4 L4-10: While these goals effectively cover key opportunities in agricultural research, item (iv) probably needs further unpacking to be more useful. Can global change be disaggregated into key components and goal/research prioritized?. For instance, in table 2, priority 4, the focus seems to be on climate change adaptation research and maybe this priority needs to be clearly stated.

P9 L9 There is recent research that would nicely complement this statement. Suggest considering the three key issues highlighted by Coe et al (2014) as determinants of large scale adoption following sentence: …Often the problem is much more complex than that..…..

Coe R., Sinclair F., Barrios E. 2014. Scaling up agroforestry requires research 'in' rather than 'for' development. Current Opinion in Environmental Sustainability 6: 73-77.

P20 Suggest the following revision in Table 2, priority 4, complexity research approaches: 'Understanding the relationship between functional diversity of soil organisms and adaptive capacity in the face of climate change'

Specific comments: P11 L19 …outbreak. This is due TO risk spreading, or… P18 L22 …were only observed in 55% of the…. P27 L1 Correct publication year to 2015.

Please also note the supplement to this comment:
http://www.soil-discuss.net/soil-2016-7/soil-2016-7-RC1-supplement.pdf

---

## Referee Comment (RC2) · P.D. Hallett (Referee) · 26 May 2016

This is a very well written and thought provoking paper on the needs of soil science and the current failures in the way we address the challenges of food security and sustainability. I think that it will raise good discussion amongst soil scientists, with some disagreement to the arguments presented in the text generating a healthy debate. Many of the examples in the text focus on subsistence farming and I think it may be better to alter the entire focus of the paper to these systems. This is where socioeconomic and political elements have a large effect on the impact of our science. Moreover, these areas have been poorly served by soil science in the past. I would argue that soil science in developed countries has become somewhat disconnected from food production. We study how much farming affects soils, but sometimes ignore how much food is being produced in our experiments. In this science we also fail to account for the economic

viability of different practices. These arguments come out to some extent in this paper, but I think the author could be more explicit about what is missing and who we should be working with.

Soil science has become the poor cousin of plant science in the push for sustainable food production. I think text should be added on how we can work more effectively with this research community. Plant genetics will not solve everything, and it is becoming increasing recognised by plant scientists that working closely with soil scientists will allow for more effective advancement. Moreover, there is increasing research effort on root systems by plant scientists, so the need for soil science input is increasing.

I liked Table 2, which summarised the research challenges succinctly. Some of this does need expanding in the text, particularly 'transdisciplinary' science. I think that economics is particularly important to go from 'soft' to 'hard' assessments of sustainability. You need to add more text on how we can fix things, rather than just on what needs to be fixed.

The referencing of past research needs improvement in this paper. I raised a couple of examples below, but the first few sentences of many paragraphs need justification. Also, avoid using popular science beliefs in making arguments. The text about GMO crops inducing toxins to soils is a good example of this. You failed to balance this with the impact on food production or farmer livelihoods if no pesticide or chemical pesticides are used.

Abstract:

This is a well written Abstract but it focusses on what you are going to discuss rather than what you have discussed in the paper, so it is more of an Introduction. Include the major conclusions from your paper here.

Lines 19 & 20 - you could combine these sentences to make the 'dimensions' clearer and decrease text length.

[Figure]

Line 22 - I am not convinced that they are entirely 'soft' concepts. There are metrics that can be used to quantify part of these dimensions. Certain instances of complexity, such as spatial variability, are possible to quantify.

Line 28 - another primary driver that should be included in the Abstract, and is included in the main body text, are socioeconomic and political drivers. Science effort tends to be addressed where production is most intensive and profitable, but it is regions with lowest incomes and political strife that could benefit considerably from our science.

Introduction

page 2, line 6 - change 'primary' to 'food'

page 2, line 11 - 'human agency' is jargon that won't be understandable to many soil scientists.

page 2, line 21 - I am not sure I totally agree. Soil scientists over the past couple of decades have become focussed on soil processes rather than on the interaction between soils and agricultural productivity. The research effort has been on what is wrong with soils, with a doom and gloom attitude of the future rather than offering solutions. We are awash with papers describing management processes that improve soil properties but would not be feasible to adopt in large scale agriculture because of resource availability (e.g. compost) or cost. Agronomic soil science has been considered to be a lower class of science, with much of the effort on enhancing food production focussing on plant genetics/breeding, with little thought about soil processes Engagement with this community of researchers is a big challenge for our discipline.

page 2, line 30 - these practices are written in a negative tone. Why be so negative about precision agriculture etc.? There is hard evidence of improved soil properties, agricultural efficiency and environmental footprints.

page 2, line 33 - I don't follow this argument, particularly for precision agriculture. It addresses sustainability directly by using fewer resources in a targeted manner. Com-

plexity and uncertainty are dealt with by sensors that identify where to apply inputs to maximise impact on the farm. With GMO it is a drive to decrease pesticide input. Yes, this is by putting toxins into the soil because these toxins kill pathogens. Pesticides kills pathogens and to some extent farmers. Pathogens kill plants, so yields are decreased and people starve.

page 3, line 6 - if the paradigms are subjective, is it not our challenge to decrease this subjectivity? System and economic analysis can help achieve this.

page 3, line 15 - is subjectivity really to blame for the example presented here? The example is socioeconomic and political. Benefits of controlled fertiliser use in SSA are understood to some extent, and they could be understood better with more research. Farmers use fertilisers in this region but they don't have access due to income. The soil fertility research in this region has not adequately addressed multiple factors, such as water availability versus fertiliser inputs.

page 3, line 29 - 'epistemological' - I had to look this word up and I am a native english speaker. You define it in brackets at the end of the sentence so remove the word and please use simpler english. Ditto for 'ontological'.

page 5, line 9 - Very nice definition of soil sustainability.

2.2 Complexity

This section would benefit from a discussion about functional redundancy in soils and the poor like between properties such as biodiversity and certain ecosystem services such as nitrogen cycling.

page 9, line 5 - it would help here to be more explicit about the need for soil scientists to engage more with socioeconomics researchers. I know you mean this in the text, but to lab based scientists who work on specialist subjects, such links may not be overtly apparent.

page 9, line 10 - this is just one example of many locations in this paper where statements need a citation.

page 11, line 32 - this addresses my comment about functional redundancy of microbial communities above to some extent. It would be clearer to the reader if you described the spatial element in greater depth, preferably with an example. A good one could be the adoption of organic farming where ecological advantage can be found at small-scale but when scaled up the resource inputs and land use can result in negative impacts.

page 13, line 9 - most of the examples used in this paper are from SSA. This is the region where the issues addressed in the paper are most pressing, so this is not a problem, but it may be better to alter the entire focus of the paper towards soil science needs to help resource poor small-holder farmers.

page 13, line 12 - 'agnostic' is not an appropriate term to use here.

page 15, line 18 - you use a toxicology rather than soil science example here. A better example for soil science would be the global adoption of ploughing of soils to depth and its widespread impact on soil degradation. After decades of use, more vulnerable soils are now farmed with lower input tillage systems, but it took a long time for adoption because of societal complacency and a lack of hard data. The Cerrado in Brazil is an excellent example.

page 23, line 7 - the transdisciplinary approach mentioned needs to be expanded. Who should soil scientists be working with, what should they work together on, and what will this achieve?

---

## Editor Comment (EC1) · P.D. Hallett (Editor) · 26 May 2016

Dear Prof Tittonell,

Many thanks for the thought-provoking paper submitted to SOILD. You will see that both reviewers provided positive comments and also suggestions for improvement. You have already addressed these for Referee 1.

It seems a major challenge in soil science is getting reviewers to conduct promised reviews, so I gave up on Reviewer 2 and did the review myself. I apologise the delay that this has caused in the review process. Can you please consider my comments as well in the revision of the manuscript?

Best regards,

[Figure]

Paul Hallett